# Route of Application and Dose Evaluation of Dental Pulp Stem Cells for the Treatment of Sialadenitis Caused by Sjögren’s Syndrome: A Preclinical Study

**DOI:** 10.3390/biomedicines13051068

**Published:** 2025-04-28

**Authors:** Zhihao Du, Lifang Feng, Yu Zhang, Xin Peng, Shan Zhang, Rui Zhao, Jia Lei, Xiaotong Li, Guangyan Yu, Chong Ding

**Affiliations:** 1Department of Oral and Maxillofacial Surgery, Peking University School and Hospital of Stomatology, No. 22, Zhongguancun South Avenue, Haidian District, Beijing 100081, China; du_zhi_hao@bjmu.edu.cn (Z.D.); pkumisamisa@163.com (Y.Z.); pxpengxin@263.net (X.P.); 2Center Laboratory, Peking University School and Hospital of Stomatology & National Center for Stomatology & National Clinical Research Center for Oral Diseases & National Engineering Research Center of Oral Biomaterials and Digital Medical Devices & Beijing Key Laboratory of Digital Stomatology & NHC Key Laboratory of Digital Stomatology & NMPA Key Laboratory for Dental Materials, No. 22, Zhongguancun South Avenue, Haidian District, Beijing 100081, China; amile_amy@sina.com (S.Z.); zr@bjmu.edu.cn (R.Z.); leijia@bjmu.edu.cn (J.L.); lixiaotong@bjmu.edu.cn (X.L.); 3Department of Prosthodontics, North China University of Science and Technology, Stomatology, Tangshan Bay Ecological Zone No. 21 Bohai Avenue, Tangshan 063210, China; fang_130cn@yeah.net

**Keywords:** Sjögren’s syndrome, stem cell therapy, dental pulp stem cells, treatment strategies, precision medicine

## Abstract

**Background:** Sjögren’s syndrome (SS) is an autoimmune disorder characterized by sicca syndrome and/or systemic manifestations. In this study, non-obese diabetic (*NOD*) mice were used as an animal model for studying SS, to evaluate the optimal administration route and dose range of dental pulp stem cells (DPSCs) in the treatment of sialadenitis caused by SS. **Methods:** Different doses of DPSCs were transplanted into the submandibular glands (SMGs) of 14-week-old *NOD* mice through two different methods: injection or retrograde perfusion through the catheter orifice into the SMG. At 21 weeks of age, the saliva flow rate (SFR), ectopic lymphocytes, and CD4^+^ T-cell infiltration were measured. Tumor necrosis factor-alpha (TNF-α) and interferon-gamma (IFN-γ) in the glandular tissues were also quantitatively detected. **Results:** Compared with untreated and PBS-injected controls, different-dose groups of the two administration methods showed an increased saliva flow rate of *NOD* mice to varying degrees, reduced infiltration of lymphocytes and CD4^+^ T cells in the SMG, and decreased IFN-γ/TNF-α levels. Finally, we compared these two administration routes and found that the perfusion of 2 × 10^5^ DPSCs presents good therapeutic effects. **Conclusions:** DPSC perfusion through the catheter orifice is a simple and effective treatment method, which is worthy of further investigation through clinical trials.

## 1. Introduction

Sjögren’s syndrome (SS) is a chronic autoimmune disease characterized by infiltration of lymphocytes into exocrine glands. Clinically, 95% of SS patients have dry eyes and a dry mouth due to the ectopic infiltration of lymphocytes into the lacrimal and salivary glands. This dryness can last for more than 10 years, severely affecting patients’ quality of life. Additionally, approximately 60% of patients also have other autoimmune disorders, including rheumatoid arthritis and autoimmune thyroid disease, resulting in symptoms associated with multisystem damage [1,2,3]. The systemic extraglandular manifestations of this condition include polyarthralgia/polyarthritis, cutaneous vasculitis, bronchiolitis/pneumonia, tubulointerstitial nephritis, and peripheral neuropathy, as well as an increased risk of lymphoma [4]. These co-existing conditions can even be life-threatening.

Currently, the treatment of SS is still empirical. The available approaches focus mainly on relieving symptoms and preventing complications, and they are based on the treatment approaches used for other autoimmune diseases. Corticosteroids combined with immunosuppressants such as hydroxychloroquine is the first-line clinical treatment [5,6]. However, long-term use of glucocorticoids may lead to osteoporosis, hyperglycemia, central obesity, and other metabolic abnormalities.

Recent advances in the understanding of SS pathogenesis have prompted researchers to use new biological therapies. For example, B-cell-targeted therapies and T-cell-targeted therapies can be used. Rituximab is an anti-CD20 B lymphocyte depletion agent that has therapeutic effects on SS-related systemic manifestations (parotid gland swelling, lymphoma, and inflammatory arthritis) [5,7,8] and sicca manifestations (dry mouth) [7,9,10]. However, in other randomized controlled trials, insufficient evidence was found that rituximab improved dry mouth symptoms in SS patients [11,12]. T-cell-targeting medications include abatacept and alefacept, and cytokine-targeting agents include infliximab, tofacitinib, and IL-2 [13,14,15]. RO5459072, a cathepsin S inhibitor, was shown to attenuate CD4^+^ T-cell activation, but it did not confer a clinical benefit in SS patients [16]. These studies revealed that regardless of whether B-cell targeting or T-cell targeting is used, the therapeutic effect is nonselective and has limited benefits for exocrine function and sicca symptoms. Thus, identifying a therapeutic agent that targets CD4^+^ T cells in SS-damaged glands could significantly improve disease management.

Cell therapy, which involves the use of regenerative medicines designed to restore the function of damaged tissue, has gained significant traction for SS treatment. In previous studies, bone marrow-derived mesenchymal stem cells (BMMSCs) [17], umbilical cord-derived mesenchymal stem cells (UCMSCs) [18], and labial gland-derived mesenchymal stem cells (LGMSCs) [19] were used in non-obese diabetic (*NOD*) mice, a classic animal model with SS-like features. After administration of these cells, the saliva flow rate (SFR) improved, and glandular ectopic lymphocyte infiltration decreased. Compared with these MSCs, dental pulp stem cells (DPSCs) and stem cells derived from exfoliated deciduous teeth (SHEDs) originate from unique stem cell sources and are obtained using non-invasive techniques. These cells possess the remarkable ability to undergo multidirectional differentiation, including differentiation into adipocytes, odontoblasts, chondrocytes, osteoblasts, hepatocytes, neuronal cells, and myocytes [20,21,22,23,24], and they can exert immunoregulatory effects [25,26]. In a previous study, intravenous injection of SHEDs was shown to exert protective effects on saliva secretion in *NOD* mice by modulating the differentiation of CD4^+^ T cells and improving the inflammatory microenvironment [27]. Additionally, DPSCs are beneficial in the treatment of chronic inflammatory disease (traumatic brain injury, osteoarthritis) [28,29] and immunological diseases (rheumatoid arthritis, systemic lupus erythematosus) [30,31], which have shown promising results. However, there are no reports on the use of DPSCs for SS treatment. Whether DPSCs can target CD4^+^ T cells to treat sialadenitis caused by SS requires further investigation. Additionally, the route of administration and the optimal dose range of these cells should be established in a preclinical study.

Therefore, this preclinical study aimed to evaluate treatment parameters and outcomes for the administration of DPSCs to the submandibular gland (SMG) for the treatment of sialadenitis caused by SS.

## 2. Materials and Methods

### 2.1. Cell Culture

DPSCs were sourced from Beijing SH Biotechnology (Beijing, China), which were isolated and cultured with the established methods [32,33]. The reagents used for cell culture were from Gibco (Rockville, MD, USA). DPSCs from passages 2–7 were used to ensure reliability and consistency.

### 2.2. Animal Experiments

*NOD* mice are the classic animal model of SS [34,35,36]. Depending on the severity of sialadenitis, the *NOD* mice go through three stages: initial stage (the age of 7–8 weeks), early clinical stage (the age of 12–16 weeks), and late clinical stage (the age of 20–24 weeks) [37]. Correspondingly, *NOD* mice at the ages of 7, 14, and 21 weeks are commonly used to investigate SS progression [27,38,39]. In the present study, the 21-week-old *NOD* mice were used with and without DPSC treatment.

### 2.3. DPSC Treatment

This study was approved by the Peking University Institutional Review Board for the Care and Use of Laboratory Animals (LA2021581). Female *NOD* mice at 7, 14, and 21 weeks of age were purchased from Beijing Huafukang Biotechnology Co., Ltd. (Beijing, China). To minimize suffering, tribromoethanol was used to anesthetize the mice via intraperitoneal injection. *NOD* mice aged 7, 14, or 21 weeks that were not subjected to any treatments were used as an untreated group. For injection, 1 × 10^5^, 2.5 × 10^5^, 5 × 10^5^, or 1 × 10^6^ DPSCs (diluted in 20 μL of PBS) were injected directly into the SMG in 14-week-old *NOD* mice at multiple points. An equal volume of PBS was injected as a vehicle control. For intraductal perfusion, an intrathecal catheter device (32 g with 27 g needles) was inserted into the intraoral duct orifice of the SMG and used to perfuse 7.5 × 10^4^, 1.5 × 10^5^, 2 × 10^5^, 3 × 10^5^, or 4 × 10^5^ DPSCs (diluted in 20 μL of PBS). An equal volume of PBS was perfused as a vehicle control. The mice in both the injection group and the intraductal perfusion group were treated once at 14 weeks of age and sacrificed at 21 weeks of age for further experiments. For all animal experiments, there were 6 mice in each group.

### 2.4. DPSC Tracking

Before injection or perfusion, the DPSCs were stained with 1,1-dioctade-cyl-3,3,3,3-tetramethylindotricarbocyanine iodide (DiR) for 2 h for labeling. The DiR-labeled DPSCs were subsequently injected or perfused into *NOD* mice at 14 weeks of age. An in vivo imaging system (Caliper Life Sciences, Hopkinton, MA, USA) was used to observe the location of DiR-labeled DPSCs at 1 and 7 days after injection or perfusion.

### 2.5. Stimulated SFR Measurement

To stimulate salivary secretion, 21-week-old *NOD* mice were intraperitoneally injected with pilocarpine (5 μg per 10 g body weight) under anesthesia. Saliva was collected from the mouth for 10 min with a micropipette and weighed with an electronic balance.

### 2.6. Histological Evaluation of SMG

As described in our previous studies, the SMG tissues of *NOD* mice were used to prepare paraffin sections, and then hematoxylin–eosin (HE) staining was performed [27,40,41]. The lesion score (the number of lesions containing more than 50 monocytes in a 4 mm^2^ area) and lesion ratio (the ratio of the lesion area to the total tissue area) were subjected to quantitative analysis. Immunohistochemical staining was performed after the paraffin-embedded samples were deparaffinized, heat-induced antigen repair was performed, and the samples were stained with an anti-CD4 antibody (Abcam, Cambridge, MA, USA), as described previously [41]. The CD4 ratio index was defined as the ratio of CD4^+^ cells’ area to the lesion area.

### 2.7. Enzyme-Linked Immunosorbent Assay (ELISA)

SMG homogenates were collected from mice in each group. Then, TNF-α and IFN-γ levels were examined by TNF-α and IFN-γ ELISA kits (Neobioscience, Shenzhen, China) according to the manufacturer’s instructions.

### 2.8. Statistical Analysis

The experimental data are presented as the means ± standard deviations. Statistical analysis was performed via Student’s *t* test for comparisons between two groups or one-way analysis of variance followed by Bonferroni correction for comparisons among multiple groups using GraphPad Prism 8.0 software. A *p* value < 0.05 was regarded as indicating statistical significance.

## 3. Results

### 3.1. Injection of DPSCs into the SMG Promotes Saliva Secretion and Ameliorates Ectopic Lymphocyte Infiltration into the Glandular Tissues of NOD Mice

As defined cellular characteristics are critical for the clinical use of stem cells, we explored the phenotype of DPSCs and found that they expressed characteristic MSC markers, including CD44, CD73, and CD90 (Appendix A). To confirm the effect of DPSCs on SS-induced sialadenitis, we injected DPSCs into the SMG of *NOD* mice at 14 weeks of age and assessed the SFR and the degree of lymphatic infiltration at 21 weeks (Figure 1A). Initially, positive signals of DiR-labeled DPSCs were observed around the neck on the 1st and 7th days post-injection (Figure 1B, left and middle). Further observation on the 7th day after injection revealed that the positive signals were present in the SMG but not in other tissues (Figure 1B, right).

Compared with that in 7-week-old *NOD* mice, the SFR was significantly lower in 14-week-old *NOD* mice and further decreased in 21-week-old *NOD* mice in the untreated group. In the treated groups, injection of 1 × 10^5^ DPSCs did not change the SFR compared with that in the PBS-treated group. Compared with the vehicle control and injection of 1 × 10^5^ DPSCs, injection of 2.5 × 10^5^, 5 × 10^5^, or 1 × 10^6^ DPSCs significantly increased the SFR (Figure 1C).

Salivary gland epithelial cells are the main targets of immune attack. Ectopic lymphocyte infiltration into the salivary glands plays a crucial role in the development of hypersalivation caused by SS [42,43,44]. Histopathological evaluation revealed that lymphocyte infiltration foci appeared in the SMG of 14-week-old mice and were aggravated at 21 weeks. In the age-matched untreated control, vehicle control, and 1 × 10^5^ and 2.5 × 10^5^ DPSC-treated groups, there were still obvious lymphocyte infiltration foci in the glandular tissues. However, in the 5 × 10^5^ and 1 × 10^6^ DPSC-treated groups, the ectopic infiltration of lymphocytes was clearly alleviated (Figure 1D). Quantitative analysis further verified that both the lesion score and lesion ratio were increased in 14-week-old *NOD* mice and further increased at 21 weeks. Compared with the untreated and vehicle control groups, the groups injected with 5 × 10^5^ and 1 × 10^6^ DPSCs presented significantly lower lesion scores and lesion ratios (Figure 1E and Figure 1F, respectively).

### 3.2. Injection of DPSCs into the SMG Decreases CD4^+^ Cell Infiltration and the Levels of SS-Specific Inflammatory Cytokines in the Glandular Tissues of NOD Mice

Excessive activation of CD4^+^ T cells is a common hallmark of autoimmune diseases such as systemic lupus erythematosus and SS [44,45]. As shown in Figure 2A,B, the ratio of the area of positively stained CD4^+^ T cells to the total area was greater in the SMG of 14- and 21-week-old *NOD* mice than in that of 7-week-old *NOD* mice. Compared with the vehicle control, injection of 2.5 × 10^5^, 5 × 10^5^, and 1 × 10^6^ DPSCs significantly decreased the ratio of the CD4^+^ T-cell area to the total area. Compared with the groups injected with 1 × 10^5^ and 2.5 × 10^5^ DPSCs, the groups injected with 5 × 10^5^ and 1 × 10^6^ DPSCs presented decreased ratios of CD4^+^ T-cell area to the total area.

TNF-α and IFN-γ are crucial inflammatory cytokines that are highly expressed in the periphery and damaged glands of SS patients and *NOD* mice [46,47,48]. Compared with those in 7-week-old *NOD* mice, TNF-α levels were lower in the 2.5 × 10^5^ DPSCs group and further decreased in the 5 × 10^5^ and 1 × 10^6^ DPSC groups. The IFN-γ content in the SMG was increased in 21-week-old *NOD* mice. Compared with the vehicle control, injection of 5 × 10^5^ or 1 × 10^6^ DPSCs resulted in lower levels of IFN-γ. In particular, injection of 5 × 10^5^ DPSCs further decreased IFN-γ levels compared with those observed in the 1 × 10^5^ DPSC group and the 2.5 × 10^5^ DPSC group (Figure 2C, D). The above results are summarized in Table 1.

These data are consistent with the observed changes in the SFR and suggest that DPSC injection has a therapeutic effect on hyposalivation caused by SS. In particular, injection of 5 × 10^5^ DPSCs appears to be the optimal dose.

### 3.3. Intraductal Perfusion of DPSCs into the SMG Restores Saliva Secretion and Ameliorates Ectopic Lymphocyte Infiltration into the Glandular Tissues of NOD Mice

Stem cell-based cell therapies are typically delivered via intravenous or topical administration. For treatment of the salivary glands, intraductal perfusion is undoubtedly a better approach. Therefore, in parallel, we perfused DPSCs through the orifice of the SMG duct in 14-week-old *NOD* mice and detected the perfused cells at 21 weeks (Figure 3A). After perfusion, a DiR-DPSC-positive signal was detected around the mouth area on the 1st day and was then captured around the neck on the 7th day (Figure 3B left and middle). Further observation on the 7th day revealed that the positive signals were present in the SMG but not in other tissues (Figure 3B right). These results indicate that the perfused DPSCs passed through the ducts and migrated into the glandular tissue.

Compared with the vehicle control, perfusion of 7.5 × 10^4^ DPSCs did not change the SFR. As expected, 1.5 × 10^5^, 2 × 10^5^, 3 × 10^5^, or 4 × 10^5^ DPSC perfusion increased the SFR in 21-week-old *NOD* mice compared with the mice in the age-matched untreated group, vehicle control group, and 7.5 × 10^4^ DPSC perfusion group (Figure 3C). Similarly, perfusion of 7.5 × 10^4^ DPSCs did not suppress ectopic lymphocytic infiltration, whereas perfusion of 1.5 × 10^5^, 2 × 10^5^, 3 × 10^5^, or 4 × 10^5^ DPSCs significantly reduced both the lesion score and the lesion ratio in the SMG compared with those in the untreated group, vehicle control group, and 7.5 × 10^4^ DPSC perfusion group (Figure 3D–F).

CD4^+^ T-cell staining revealed that administration of 2 × 10^5^, 3 × 10^5^, and 4 × 10^5^ DPSCs reduced the ratio of the CD4^+^ T-cell area to the total area in the SMG compared with that in the untreated, vehicle control, and 7.5 × 10^4^ DPSC groups (Figure 4A,B). Furthermore, compared with perfusion of the vehicle control, perfusion of 1.5 × 10^5^ or 2 × 10^5^ DPSCs decreased TNF-α levels. Compared with perfusion of 7.5 × 10^4^ DPSCs, perfusion of 1.5 × 10^5^ DPSCs reduced TNF-α levels (Figure 4C). Perfusion of both 2 × 10^5^ and 3 × 10^5^ DPSCs decreased IFN-γ levels compared with those in the vehicle control and 7.5 × 10^4^ DPSC groups (Figure 4D). The above results are summarized in Table 2.

These results indicate that intraductal perfusion of DPSCs is indeed a good administration option for the treatment of SS-induced hyposalivation. The optimal dose for DPSC injection is 5 × 10^5^ cells.

### 3.4. Comparison of the Two Application Routes

Based on the above data, we surmised that the optimal dose of DPSC injection was 5 × 10^5^, and the optimal dose of intraductal perfusion was 2 × 10^5^. To reveal their therapeutic efficacy, we next compared these two application routes (Figure 5A). We found that 5×10^5^ injection and 2 × 10^5^ perfusion both promoted the SFR by 84% and 63% compared their vehicle control, respectively. And there was no difference between these two treated groups (57.07 ± 11.42 vs. 55.48 ± 3.87, *p* = 0.75; Figure 5B). Further, both administration methods can reduce the focus score and the ratio index in SMGs. The perfusion group showed the same inhibitory effect on the focus score (2.33 ± 0.52 vs. 1.67 ± 0.82, *p* = 0.12, Figure 5C). However, the ratio index of ectopic inflammatory infiltration notably declined in the 2 × 10^5^ perfusion group compared to the 5 × 10^5^ injection group (0.13 ± 0.02 vs. 0.07 ± 0.01, *p* < 0.01, Figure 5D). On the contrary, the 5 × 10^5^ injection group showed a better inhibiting effect on the CD4^+^ T-cell-positive staining ratio (0.24 ± 0.05 vs. 0.37 ± 0.05, *p* < 0.01, Figure 5E). The TNF-α level shown the same reduced effect in the two groups (13.00 ± 7.09 vs. 21.57 ± 12.46, *p* = 0.27, Figure 5F). However, the 5 × 10^5^ injection group showed the better inhibiting effect on IFN-γ (495.2 ± 113.83 vs. 767.2 ± 254.16, *p* = 0.04, Figure 5F). The above results are summarized in Table 3. Collectively, these findings suggest that 2 × 10^5^ DPSC perfusion is best suited for clinical application.

## 4. Discussion

In this article, we present preclinical results regarding the use of DPSCs as a potential cell therapy for sialadenitis caused by SS. We performed preclinical in vivo assessments of safety and therapeutic efficacy and determined the optimal route of administration and cell dose in an SS mouse model.

DPSCs were first derived from the third molars of humans by Gronthos in 2000 [20], and they can also be derived from orthodontic teeth and supernumerary teeth. They possess a stable self-renewal ability and multidirectional differentiation potential [49,50]. Compared with other types of MSCs, DPSCs exhibit stronger proliferative, anti-inflammatory, and antifibrotic effects [51,52] and do not exhibit tumor formation after transplantation like embryonic stem cells (ESCs) or induced pluripotent stem cells (iPSCs) [53]. These characteristics make DPSCs valuable for disease treatment and regenerative therapy [54,55]. In detail, DPSCs can exert immunosuppressive effects by interfering with activated T cells [56] and exert anti-inflammatory effects by inducing M2 polarization in macrophages [57]. Recent reports indicate that the secreted molecules within DPSCs contain diverse trophic factors and demonstrate efficacy in multiple disease models [58,59,60,61]. More importantly, cytokine production of DPSCs outperforms BMMSCs in reducing inflammation, promoting tissue repair, and combating infection [62]. In the present study, we confirmed that DPSCs exert therapeutic effects, promoting saliva production and alleviating inflammation in glandular tissues in *NOD* mice, via both injection and perfusion into the SMG. After injection or perfusion, the DPSCs were confined to glandular tissues, indicating the safety of local administration.

During the pathological process associated with SS, ectopic lymphocyte infiltration (mostly infiltration of CD4^+^ T cells) occurs; this is the first immune defense against injury or infection [63]. If the inflammatory process lasts too long, many activated cells are attracted to the site of injury, releasing excessive amounts of enzymes, chemokines, etc., which is detrimental to tissue repair and promotes further deterioration [64]. In this study, we evaluated the inflammatory condition of the SMG in *NOD* mice at 7, 14, and 21 weeks and found that, over time, the disease progressed, with intensified ectopic lymphatic infiltration, which was consistent with our previous studies [27,39]. However, after DPSC treatment, both the lesion score and the lesion ratio were related to lymphocyte infiltration, and the number of infiltrating CD4^+^ T cells was effectively reduced.

Additionally, the current consensus regarding SS pathogenesis is that diverse environmental factors act on susceptible individuals, leading to the disruption of autoimmune tolerance. Activated CD4^+^ T cells can secrete IL-2, TNF-α, IFN-γ, IL-10, and other inflammatory factors. Among these factors, TNF-α and IFN-γ are the most typical, and the levels of both are increased in SS-damaged tissues [27,65]. Single-cell RNA sequencing in combination with pathway enrichment analysis of peripheral blood mononuclear cells demonstrated the upregulation of genes associated with type I and II interferon signaling, TNF family signaling, and antigen processing and presentation processes in SS patients [66]. TNF-α can affect the local inflammatory immune microenvironment by stimulating the secretion of other proinflammatory factors and participating in the disruption of the tight junctions of glandular epithelial cells [48]. It can also increase the expression of two important autoantigens associated with SS, Ro/SSA and La/SSB [67], and may cause secretory dysfunction in SS by inducing apoptosis in glandular cells alone and with IFN-γ [68]. On the other hand, IFN-γ can induce the expression of adhesion molecules, causing a large number of inflammatory cells to aggregate and initiating local inflammatory damage [69]. Patients with the cluster characterized by the most robust IFN and inflammation gene signatures exhibited elevated ESSDAI scores, along with increased levels of anti-Ro/SSA and La/SSB autoantibodies [70]. FN-γ can also activate the innate immune system, increase the release of proinflammatory factors, increase antigen presentation, and damage the tight junction structure of salivary gland epithelial cells, resulting in persistent dysfunction of glandular secretion [71]. Previously, we verified that the expression of TNF-α and IFN-γ in 21-week-old *NOD* mice significantly increased with age. SHED administration via the tail vein can decrease TNF-α and IFN-γ levels in serum, SMG tissues, and saliva and, ultimately, alleviate hyposalivation caused by SS [27]. Here, after DPSC administration, both TNF-α and IFN-γ levels in the SMG were significantly reduced, which was consistent with the ectopic lymphatic infiltration and CD4^+^ T-cell staining results. Collectively, these results confirm that DPSC administration exerts a therapeutic effect on SS sialadenitis through the immunosuppression of CD4^+^ T cells.

Finally, compared with injection via the tail vein or direct injection into glandular tissue, intraductal perfusion is a good route of administration for clinical applications because the catheter system can guide reverse transport of DPSCs into the glandular tissue without additional damage to the glandular lobe. Previously, we performed intraductal perfusion of SHED exosomes into the SMG in 14-week-old *NOD* mice and confirmed the ability of the administered exosomes to increase the SFR and alleviate inflammatory infiltration [40]. In the present study, we further attempted to perform intraductal perfusion of DPSCs into the SMG and proved that DPSC perfusion has a therapeutic effect superior to that of injection, increasing the SFR and ameliorating the inflammatory response in SS. Furthermore, in terms of the translational feasibility, DPSC perfusion has the advantage of non-invasive administration, meaning it is more easily accepted by patients and doctors in clinical practice.

In future, the mechanism of DPSCs (whether DPSCs can interact with immune cells or glandular epithelium) requires further in-depth research. Further, the human disease pathology exhibits more complexity, mandating rigorous validation of pharmacokinetic profiles in humanized models, and the long-term efficacy and safety profiles also need further evaluation.

## 5. Conclusions

In the present study, various doses of DPSCs and the optimal route of administration for the treatment of sialadenitis caused by SS were assessed. Safety and efficacy tests were performed in this preclinical study. Both injection and perfusion of DPSCs into the SMG at various doses exhibited therapeutic effects in *NOD* mice via the targeting of CD4^+^ T cells. The perfusion of 2 × 10^5^ DPSCs is a convenient and effective therapeutic method. Overall, these findings provide a vital basis for designing experiments to optimize the route of administration and dose of DPSCs in clinical trials.

## Figures and Tables

**Figure 1 biomedicines-13-01068-f001:**
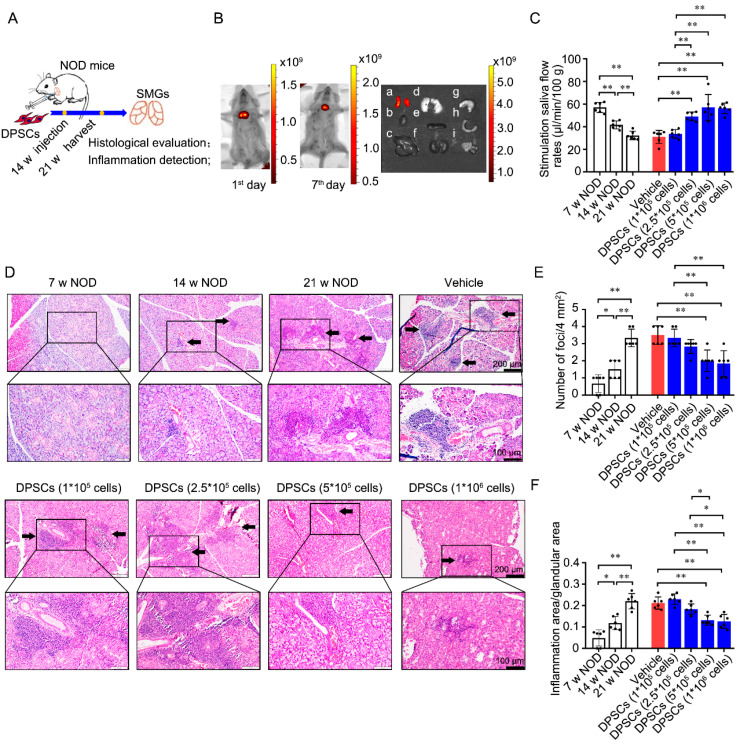
DPSCs injected into the SMGs promote saliva secretion and ameliorate ectopic lymphocyte infiltration into glandular tissues of *NOD* mice. (**A**) Scheme of animal experiments. Different concentrations (1 × 10^5^, 2.5 × 10^5^, 5 × 10^5^, 1 × 10^6^) of DPSCs were injected into the SMGs of 14-week-old *NOD*. Further analyses were performed at 21 weeks. (**B**) Bioluminescence imaging was performed on the 1st and 7th days post-injection of DiR-DPSCs. Then, other organs were harvested on the 7th day. a: SMG; b: heart; c: liver; d: lung; e: spleen; f: kidney; g: stomach; h: intestines; i: pancreatic tissue. (**C**) Stimulated SFR of 7-, 14-, and 21-week-old *NOD* mice with or without DPSC injection. (**D**) Representative histological images of SMGs in above eight groups. Arrows indicate lymphocyte foci. (**E**,**F**) The degree of inflammatory infiltration in SMGs, assessed by focus score and ratio index. n = 6. * *p* < 0.05 and ** *p* < 0.01.

**Figure 2 biomedicines-13-01068-f002:**
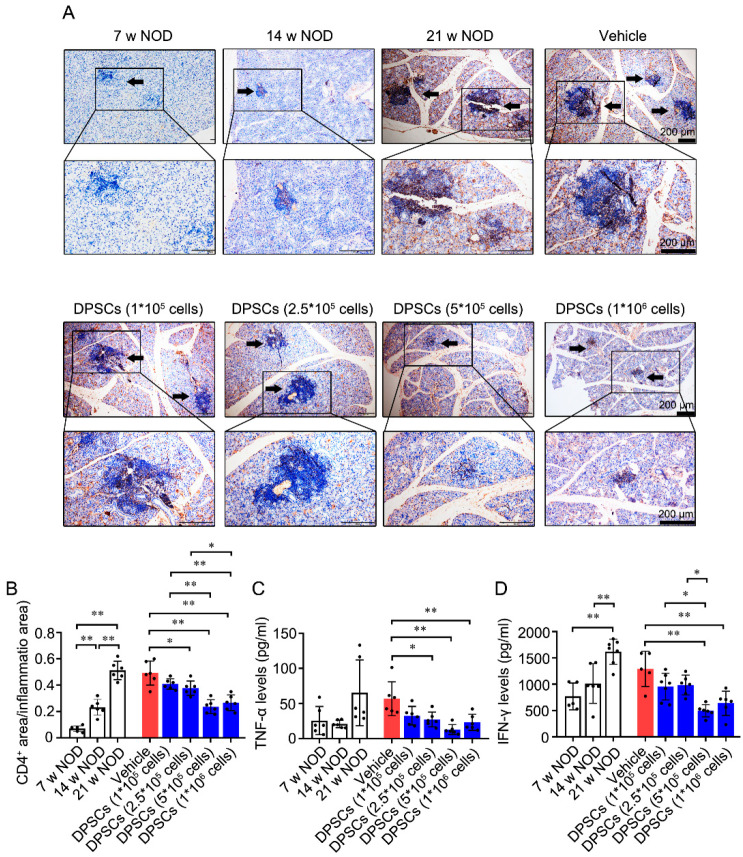
DPSCs injected into the SMGs decrease the CD4^+^ cell infiltration and SS-specific inflammatory cytokines in glandular tissues of *NOD* mice. (**A**) Representative immunohistochemical staining images of CD4^+^ in SMG tissues across the eight groups. Arrows indicate CD4^+^ cell foci. (**B**) Quantitative immunohistochemical analysis. (**C**,**D**) IFN-γ and TNF-α levels in SMG tissues in above eight groups. n = 6. * *p* < 0.05 and ** *p* < 0.01.

**Figure 3 biomedicines-13-01068-f003:**
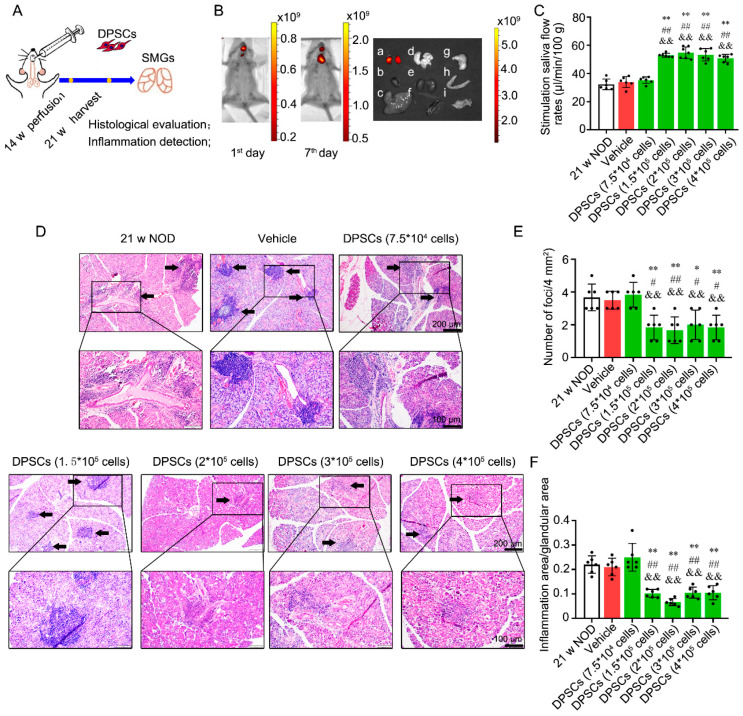
DPSC intraductal perfusion increases saliva secretion and ameliorates ectopic lymphocyte infiltration into SMGs of *NOD* mice. (**A**) Scheme of animal experiments. Different concentrations (7.5 × 10^4^, 1.5 × 10^5^, 2 × 10^5^, 3 × 10^5^, 4 × 10^5^) of DPSCs were intraductally perfused into the SMGs of 14-week-old *NOD*. Further analyses were performed at 21 weeks. (**B**) Bioluminescence imaging was performed on the 1st and 7th days post-injection of DiR-DPSCs. Then, organs were harvested and detected on the 7th day. a: SMG; b: heart; c: liver; d: lung; e: spleen; f: kidney; g: stomach; h: intestines; i: pancreatic tissue. (**C**) Stimulated SFR of 21-week-old *NOD* mice with or without DPSC perfusion. (**D**) Representative histological images of SMGs in these seven groups. Arrows indicate inflammatory cell foci. (**E**,**F**) The degree of inflammatory infiltration into gland tissues, assessed by focus score and ratio index. n = 6. * *p* < 0.05, ** *p* < 0.01 compared with 21 w untreated *NOD* group. # *p* < 0.05, ## *p* < 0.01 compared with vehicle group. && *p* < 0.01 compared with 7.5 × 10^4^ DPSC infusion group.

**Figure 4 biomedicines-13-01068-f004:**
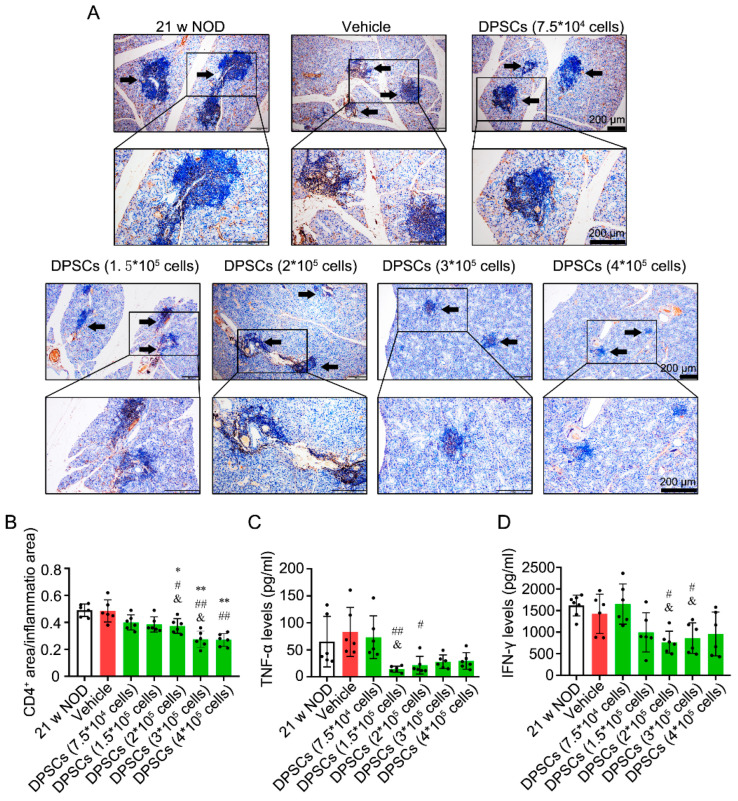
DPSC perfusion into the SMGs decreases the CD4^+^ T-cell infiltration and TNF-α/IFN-γ levels in SMGs of *NOD* mice. (**A**) Representative immunohistochemical staining images of CD4^+^ in SMG tissues in above eight groups. Arrows indicate CD4^+^ cell foci. (**B**) Quantitative immunohistochemical analysis. (**C**,**D**) IFN-γ and TNF-α levels in SMG tissues in above eight groups. n = 6. * *p* < 0.05, ** *p* < 0.01 compared with 21 w untreated *NOD* group; # *p* < 0.05, ## *p* < 0.01 compared with vehicle group; & *p* < 0.05 compared with 7.5 × 10^4^ DPSC infusion group.

**Figure 5 biomedicines-13-01068-f005:**
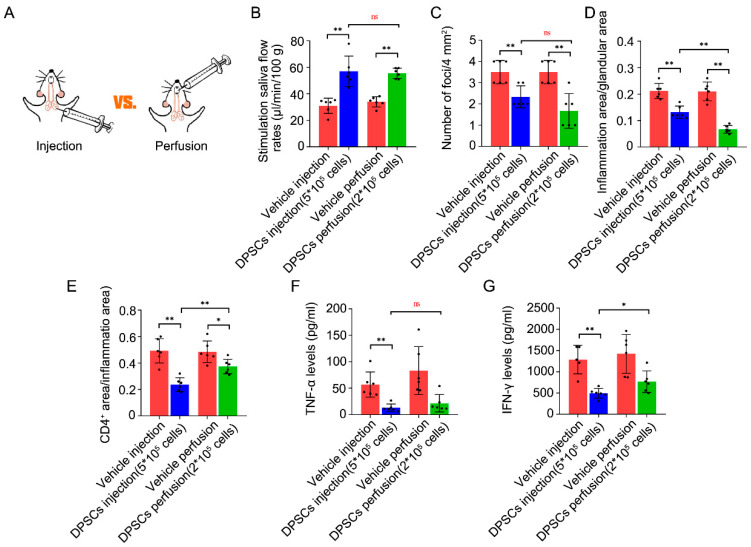
Comparison of the two administration methods. (**A**) 5 × 10^5^ DPSC injection and 3 × 10^5^ DPSC perfusion were chosen to compare their therapeutic effects for hyposalivation caused by SS. (**B**) The stimulated SFR at 21 weeks. (**C**,**D**) The focus score and ratio index in SMG tissues. (**E**) Quantitative immunohistochemical analysis of CD4^+^ area ratio in SMG tissues. (**F**,**G**) IFN-γ and TNF-α levels in SMG tissues. Values represent the mean ± SD from 6 independent experiments. * *p* < 0.05, ** *p* < 0.01, ns means no significance.

**Table 1 biomedicines-13-01068-t001:** The therapeutic effect of DPSC injection into the SMG in *NOD* mice (compared with vehicle control group).

DPSC Injection	1 × 10^5^ Cells	2.5 × 10^5^ Cells	5 × 10^5^ Cells	1 × 10^6^ Cells
Increase the stimulated SFR	No	Yes	Yes	Yes
Decrease the lesion score	No	No	Yes	Yes
Decrease the lesion ratio	No	No	Yes	Yes
Alleviate the infiltration of CD4^+^ cells	No	Yes	Yes	Yes
Reduce the level of TNF-α	No	Yes	Yes	Yes
Reduce the level of IFN-γ	No	No	Yes	Yes

**Table 2 biomedicines-13-01068-t002:** The therapeutic effect of DPSC perfusion into the SMG in *NOD* mice (compared with vehicle control group).

DPSC Perfusion	7.5 × 10^4^ Cells	1 × 10^5^ Cells	2 × 10^5^ Cells	3 × 10^5^ Cells	4 × 10^5^ Cells
Increase the stimulated SFR	No	Yes	Yes	Yes	Yes
Decrease the lesion score	No	Yes	Yes	Yes	Yes
Decrease the lesion ratio	No	Yes	Yes	Yes	Yes
Alleviate the infiltration of CD4^+^ cells	No	No	Yes	Yes	Yes
Reduce the level of TNF-α	No	Yes	Yes	No	No
Reduce the level of IFN-γ	No	No	Yes	Yes	No

**Table 3 biomedicines-13-01068-t003:** Comparison the therapeutic effects of the two application routes.

	5 × 10^5^ injection vs. 2 × 10^5^ perfusion
Application routes’ advantage	2 × 10^5^ perfusion is non-invasive
Increase the stimulated SFR	No statistical difference
Decrease the lesion score	No statistical difference
Decrease the lesion ratio	2 × 10^5^ perfusion is better
Alleviate the infiltration of CD4^+^ cells	5 × 10^5^ injection is better
Reduce the level of TNF-α	No statistical difference
Reduce the level of IFN-γ	5 × 10^5^ injection is better

## Data Availability

All data generated or analyzed in this study are included in this published article. Any other data will be available on reasonable request from the corresponding author.

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
