# Peer review of "Route of Application and Dose Evaluation of Dental Pulp Stem Cells for the Treatment of Sialadenitis Caused by Sjögren’s Syndrome: A Preclinical Study"

_biomedicines, 2025, doi:10.3390/biomedicines13051068_

Round 1

Reviewer 1 Report

Comments and Suggestions for Authors

This is a valuable manuscript. My suggestions to further improve it are as follows:

Abstract: No headings should be use in the abstract. The description of the methods in the abstract part is incomplete. The results are not understandable unless the information in the methods part is completed. The conclusion part of the abstract should be brief. 

Introduction: Overall is clear and puts the subject into perspective. Lines 99-103 should be removed or relocated as conclusions.

Materials and methods: The groups and the treatment used for each group are not clear. I suggest tabeling these informations and labeling each group. If only 14 weeks mice were treated, why 7, 14, and 21 weeks of age mice were purchased and mice aged 7, 14, or 21 weeks that were not subjected to any treatments were used as untreated group. Please clarify this matter.

Results: In my opinion the results are difficult to follow, tabeling and summarizing them would be beneficial. 

Discussion: Please verify if full term for ESCs or iPSCs was previously given, if not please give it at this point. 

Line 307-311. Please avoid repeating information already given in the introduction.

The discussion is mainly focused on the author's previous studies. A wider  perspective on related literature on this subject would be beneficial. 

Conclusion: A conclusion part is mandatory, numbered as a distinct section of the manuscript. Please add it.

The references should follow the MDPI style.

Author Response

Thank you very much for taking the time to review this manuscript. Please see the attachment with the detailed responses. Thank you very much!

Reviewer 2 Report

Comments and Suggestions for Authors

The manuscript titled “Route of Application and Dose Evaluation of Dental Pulp Stem Cells for the Treatment of Sialadenitis Caused by Sjögren’s Syndrome: A Preclinical Study” provides a well-structured and insightful investigation into the therapeutic potential of DPSCs in a relevant autoimmune model. The study’s strength lies in its dual approach—comparing injection and intraductal perfusion routes—to determine the most effective DPSC dose for ameliorating glandular inflammation and restoring salivary function in NOD mice. The robust experimental design includes histological, immunological, and functional evaluations. However, the authors could enhance the clinical relevance of their findings by contextualizing them within recent advances in DPSC-based therapies, especially studies addressing their immunomodulatory effects and regenerative capacity, such as those reported in PMID: 40164523 and PMID: 39068091

While the data convincingly show that DPSCs alleviate inflammation and increase saliva secretion via CD4+ T cell suppression and cytokine downregulation, some areas warrant clarification. First, the rationale behind selecting specific doses and timing for cell administration could be better explained. Second, although both delivery routes showed efficacy, the manuscript would benefit from a more detailed comparison of their translational feasibility. It would also be valuable to address how the findings align with existing literature on DPSC therapy for other inflammatory and autoimmune diseases. A mechanistic discussion on how DPSCs interact with resident immune cells or glandular epithelium could strengthen the interpretation.

Finally, a key limitation of the study is the lack of functional validation beyond cytokine suppression, such as proteomic or transcriptomic profiling to support the immunomodulatory role of DPSCs in SS. A brief comment on this and future directions involving humanized models or co-therapy approaches would add depth. The authors are also encouraged to refine the language in several sections, as minor grammatical inconsistencies and sentence construction issues may affect readability. Nonetheless, with these improvements, the manuscript offers a meaningful contribution to the growing field of stem cell-based therapies for autoimmune diseases.

Author Response

(The authors gave the same response as above.)

Round 2

Reviewer 1 Report

Comments and Suggestions for Authors

Apart from some minor revisions, the authors didn't perform the suggested modifications.  The results and discussion sections were not revised, despite the fact that the authors claim they did it (see response to the reviewers, comment 4 and 7). If the authors have a divergent opinion about ammending the results and discussion part (they don't consider revisions are needed), they should argue their position on this matter, not claim that revisions were made. 

Author Response

Thank you for your suggestions and comments. We apologize for the inconvenience caused by our previous reply. Please see the attachment.

Round 3

Reviewer 1 Report

Comments and Suggestions for Authors

I have no further comments.